# RobIR: Robust Inverse Rendering for High-Illumination Scenes

**Ziyi Yang**[1]    **Yanzhen Chen**[1]    **Xinyu Gao**[1]    **Yazhen Yuan**[2]
**Yu Wu**[2]    **Xiaowei Zhou**[1]    **Xiaogang Jin**[1†]

[1]State Key Lab of CAD&CG, Zhejiang University    [2]Tencent

## Abstract

Implicit representation has opened up new possibilities for inverse rendering. However, existing implicit neural inverse rendering methods struggle to handle strongly illuminated scenes with significant shadows and slight reflections. The existence of shadows and reflections can lead to an inaccurate understanding of the scene, making precise factorization difficult. To this end, we present *RobIR*, an implicit inverse rendering approach that uses ACES tone mapping and regularized visibility estimation to reconstruct accurate BRDF of the object. By accurately modeling the indirect radiance field, normal, visibility, and direct light simultaneously, we are able to accurately decouple environment lighting and the object's PBR materials without imposing strict constraints on the scene. Even in high-illumination scenes with shadows and specular reflections, our method can recover high-quality albedo and roughness with no shadow interference. *RobIR* outperforms existing methods in both quantitative and qualitative evaluations. Code is available at https://github.com/ingra14m/RobIR.

## 1   Introduction

Inverse rendering, the task of extracting the geometry, materials, and lighting of a 3D scene from 2D images, is a longstanding challenge in computer graphics and computer vision. Previous methods, such as providing geometry for the entire scene [35, 46], modeling shape representation [21, 34, 52, 14] or pre-providing multiple known light information [10], have achieved plausible results using prior information. To achieve clear albedo and roughness decomposition, factors such as light obscuration, reflection, or refraction must be taken into account. Among these, hard and soft shadows are particularly challenging to eliminate, as they play a critical role not only in obtaining cleaner material but also in accurately modeling geometry and light sources. Although some data-driven approaches [22, 39] have performed plausible shadow removal at the image level, these methods are not generally applicable for inverse rendering.

Since the advent of NeRF [32], implicit representation has garnered significant interest in portraying scenes as neural radiance fields. By applying implicit neural representation to inverse rendering [3, 19, 54], plausible factorization can be achieved in simple scenes with weak light intensity. Thanks to NeRFactor [55] and its relevant work [8], which extend previous works by explicitly representing visibility, implicit inverse rendering can be improved with simple shadow removal and clear edge in albedo and roughness. Recently, InvRender [56] has taken the scene factorization problem to a new level by modeling indirect illumination, serving as the baseline in our experiment.

However, in high-illumination scenarios with strong shadows or subtle specular reflections, the current methods for implicit inverse rendering have shown limitations in accurately modeling each decomposed part for BRDF estimation. Especially, it will lead to shadow baking in albedo and roughness, thereby causing serious artifacts in relighting and other downstream applications. To

38th Conference on Neural Information Processing Systems (NeurIPS 2024).

deal with such scenes, the following challenges arise in order to obtain high-quality physically based rendering (PBR) materials.

First, previous methods for inverse rendering struggle to correctly decouple environment lighting, shadows, and the object's PBR materials. While these methods perform well in scenes with weak light intensity, where shadows and specular reflections are minimal, they struggle to accurately reconstruct BRDF of the object in scenarios with intense lighting. As shown in Fig. 4 and Fig. 5, shadow and specular reflection lead to poor albedo and messy environment map. To address the aforementioned challenge, we propose a novel approach that applies Academy Color Encoding System (ACES) [1] tone mapping [1] to nonlinearly and monotonically convert the PBR color output from the rendering equation to a range within $[0, 1]$. Specifically, we introduce a scaled parameter $\gamma$ to adjust the standard ACES tone mapping curve for specific scenes, better adapting to varying lighting conditions. Unlike previous methods, which either directly output PBR color within $[0, 1]$ [56], or convert linear PBR color outputted within $[0, 1]$ to sRGB color also lying in $[0, 1]$ [17, 26], our method can calculate PBR color over a broader value range. For areas with extremely strong or weak lighting, ACES tone mapping can reduce information loss in reconstruction through more refined contrast control, thereby better estimating BRDF without baking shadow or specular highlights.

Second, existing methods encounter difficulties in accurately modeling visibility. Typically these methods [56, 17] model the visibility field $V(\mathbf{x}, \boldsymbol{\omega})$ through a learned SDF field and sphere tracing, which takes position and view direction as inputs. However, the visibility field is not compatible with direct light modeled based on Spherical Gaussian (SG), resulting in many stubborn shadows remaining at the edges. To address this, we introduce a regularized visibility estimation (RVE) distilled from the visibility field to directly predict the visibility for each SG to achieve more accurate visibility. This technique significantly contributes to the BRDF estimation, enabling the separation of environment maps, albedo, and roughness without the baked shadows. We also apply octree tracing instead of sphere tracing to improve the precision of the visibility field modeling.

In summary, the major contributions of our work are:

- A novel scene-dependent ACES tone mapping for inverse rendering. It enables the high-quality albedo and roughness reconstruction in scenes with intense lighting and strong shadows.
- A novel regularized visibility estimation designed for direct SGs. It improves the visibility accuracy for each direct SG and reduces shadow residue, enhancing the overall BRDF quality of the ill-posed inverse rendering.
- The first neural field-based inverse rendering framework to achieve robust shadow removal in BRDF estimation under high-illumination scenes.

## 2 Related Work

### 2.1 Implicit Neural Representation

Neural rendering has gained popularity due to its ability to produce photorealistic images. Recently, NeRF [32] enables photo-realistic novel view synthesis using MLPs. It can handle complex light scattering and reconstruct high-quality scenes for downstream tasks.

Subsequent work has enhanced NeRF's efficiency in various ways, elevating it to new heights and enabling its use in other domains. Structure-based techniques [51, 13, 37, 15, 9, 12] have explored ways to improve inference or training efficiency by caching or distilling implicit neural representation into the efficient data structure. Hybrid methods [25, 27, 42, 43, 7] aim to improve the efficiency by incorporating explicit voxel-based data structures. Among them, Instant-NGP [33] achieves minute training by additionally incorporating hash encoding. In addition, some follow-up methods [36, 47, 50] are dedicated to recovering clear surfaces for scenes with complex solid objects by modeling a learnable SDF network, the value of which indicates the minimum distance between the input coordinate and surfaces in the scene.

In our work, we employ NeuS [47], an SDF-based volume rendering framework, to learn geometry priors for inverse rendering. Furthermore, drawing inspiration from PlenOctree [51], we construct an Octree tracer from the SDF to improve inference efficiency and accuracy compared to sphere tracing.

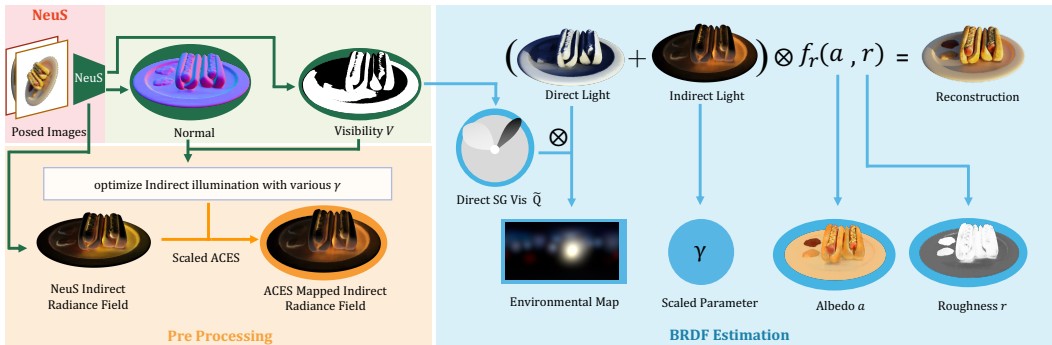

Figure 1: **The pipeline of our method.** During the pre-processing stage, we reconstruct the scene as an implicit representation by NeuS [47]. From the implicit representation, we extract scene priors such as normal, visibility, and indirect illumination. During BRDF estimation, we optimize environmental lighting, the scaled parameter $\gamma$, albedo $a$, and roughness $r$, to minimize reconstruction loss under the constraint of the rendering equation. After 100 epochs, we perform regularized visibility estimation and employ an MLP to learn the visibility ratio $\tilde{Q}$ of the direct SGs to obtain more accurate visibility specified for SGs, which is critical for eliminating stubborn shadows at the edges and boundaries.

## 2.2 Inverse Rendering

Inverse rendering is a process in computer graphics that aims to derive an understanding of the physical properties of a scene from a set of images. Because the problem is highly ill-posed, most previous works have incorporated priors such as illumination, shape, and shadow, as well as additional observations such as scanned geometry [35, 38, 20] and known light conditions [10]. Simplified approaches, such as those assuming outdoor and natural light [40] or white light [30], aim to reduce the number of fitting parameters in an ill-posed problem.

Recently, there has been a surge of interest in implicit inverse rendering, building on the success of NeRF and its fully differentiable implicit representation. To model spatially-varying bidirectional reflectance distribution function (SVBRDF) under more casual capture conditions, many recent methods [3, 19, 5, 4, 49, 53, 54] have relied on implicit representation. Other works [55, 41, 48, 17, 26] have focused on physical-based modeling for complex scenes via visibility prediction. L-Tracing [8] introduced a new algorithm for estimating visibility without training, while NeRFactor [55] proposed a canonical normal and BRDF smoothness to address NeRF's poor geometric quality. InvRender [56] extends previous work by modeling indirect illumination. Relightable-GS [11] and GS-IR [24], based on the representation of 3D-GS [18], have achieved real-time inverse rendering. However, none of these methods are able to decouple shadows and materials under high-illuminance conditions.

## 2.3 The Rendering Equation

For non-emitted object, the color $c$ of the surface point $\mathbf{x}$ is calculated by the rendering equation:

$$c(\mathbf{x}, \boldsymbol{\omega}_o) = \int_{\Omega} f_r(\boldsymbol{\omega}_o, \boldsymbol{\omega}_i, \mathbf{x}) L(\mathbf{x}, \boldsymbol{\omega}_i)(\boldsymbol{\omega}_i \cdot \mathbf{n}) d\boldsymbol{\omega}_i, \qquad (1)$$

where $c(\mathbf{x}, \omega_{\mathbf{o}})$ is the output color leaving point $\mathbf{x}$ in the view direction $\omega_{\mathbf{o}}$, $f_r(\mathbf{x}, \omega_{\mathbf{i}}, \omega_{\mathbf{o}})$ is the BRDF function, $L(\mathbf{x}, \omega_{\mathbf{i}})$ is the incoming radiance at point $\mathbf{x}$ from direction $\omega_{\mathbf{i}}$, and $\mathbf{n}$ is the surface normal. Following PhySG [54] and InvRender [56], we use spherical Gaussians (SGs) to efficiently approximate the rendering equation shown in Eq. (1). An SG is a spherical function that takes the following form:

$$G(\boldsymbol{\omega}; \boldsymbol{\xi}, \lambda, \boldsymbol{\mu}) = \boldsymbol{\mu} e^{\lambda(\boldsymbol{\omega} \cdot \boldsymbol{\xi} - 1)}, \qquad (2)$$

where $\boldsymbol{\xi} \in R^3$ is the lobe axis, $\lambda \in R^1$ is the lobe sharpness, and $\boldsymbol{\mu} \in R^3$ is the lobe amplitude. Please refer to the supplementary material for the complete details.

In NeuS [47], we can determine the surface point $\mathbf{x}$ along a specific direction using sphere tracing. By substituting the color function with the shading function based on Eq. (1), we can achieve BRDF decomposition through image loss.

# 3 Methodology

## 3.1 Overview

Given a set of multi-view RGB images with known camera poses as input, our target is to reconstruct BRDF of the object even under high-illuminance scenes. As shown in Fig. 1, the pipeline of RobIR consists of two stages. In the pre-processing stage, we train NeuS $S(x, \omega)$ as the representation of the scene, which can provide scene priors like normals, visibility, and indirect illumination (Sec. 3.2). In the BRDF estimation stage, we fix the scene priors and optimize the direct illumination and scaled parameter to compute an accurate BRDF of the object under the constraint of rendering equation (Sec. 3.3). To improve the visibility accuracy for direct illumination and decomposition stability, we introduce the regularized visibility estimation after 100 epochs (Sec. 3.4).

## 3.2 Stage 1: Pre-processing

In this stage, we adopt the same neural SDF representation and the volume rendering as NeuS [47] to reconstruct the scene. Then we can obtain the necessary prior information for the BRDF estimation stage, such as normal, visibility, and indirect illumination from NeuS.

**Normal smoothing.** In our framework, the accuracy of normal is crucial for BRDF estimation. However, we observed that normals estimated from NeuS tend to be noisy. To overcome this, we drew inspiration from Ref-NeRF [45] and employ a spatial MLP $\mathbb{N}(\mathbf{x})$ to predict smooth normals aligned with the density gradient normals (See Fig. 2) obtained from NeuS using $\mathcal{L}_2$ loss. We further employ a smooth loss to fix the broken normals caused by specular reflection:

$$\mathcal{L}_{norm} = \|\mathbb{N}(\mathbf{x}) - \hat{n}\|_2^2 + \|\mathbb{N}(\mathbf{x}) - \mathbb{N}(\mathbf{x} + \epsilon)\|_2^2, \tag{3}$$

where $\mathbb{N}$ denotes the normal at point $\mathbf{x}$ learned by MLP, $\hat{n}$ denotes the supervision normal from NeuS, and $\epsilon$ is a $0.02\times$ Gaussian noise.

**Visibility and indirect illumination.** With the availability of NeuS SDF, we can use sphere tracing to model secondary shading effects such as visibility and indirect illumination. However, performing sphere tracing requires a significant amount of time and memory. Inspired by PlenOctree [51], we use an octree tracer derived from the NeuS SDF, replacing sphere tracing to accelerate the tracing and achieve more precise intersection results. Moreover, We can further improve the inference efficiency by compressing the visibility and indirect illumination field into MLP.

As for indirect illumination, we follow InvRender [56] and model the indirect radiance field $L_I(\mathbf{x}, \boldsymbol{\omega_i})$ using $M = 24$ SGs under the supervision of NeuS radiance field. At point $\mathbf{x}$, we first perform octree tracing along direction $\omega_i$ to get the second intersection point $\hat{x}$. Then the indirect radiance field can be supervised by the out-going radiance $S(\hat{x}, -\omega_i)$ from NeuS. Then, the indirect illumination $L_I$ is computed by:

$$L_I(\mathbf{x}, \boldsymbol{\omega}; \Gamma) = \sum_{j=1}^{M} G(\boldsymbol{\omega}; \Gamma(\mathbf{x}, \gamma)), \tag{4}$$

where we use an MLP $\Gamma$ to output the $j$th indirect SG parameters, and $\gamma$ denotes the scaled parameter, which will be illustrated in Sec. 3.3.

As for visibility, we learn an MLP that maps the point $\mathbf{x}$ and direction $\boldsymbol{\omega}$ to visibility $V(\mathbf{x}, \boldsymbol{\omega})$, which is supervised by the result of octree tracer from point $\mathbf{x}$ along direction $\boldsymbol{\omega}$. The $\mathcal{L}_{indir}$ and $\mathcal{L}_{vis}$ are optimized by $\mathcal{L}_1$ and binary cross entropy loss as follows:

$$\mathcal{L}_{indir} = \|\hat{L}_I - L_I\|_1, \mathcal{L}_{vis} = \text{BCE}(V(\mathbf{x}, \boldsymbol{\omega}), \hat{V}(\mathbf{x}, \boldsymbol{\omega})), \tag{5}$$

where $\text{BCE}(p_i \| y_i)$ represents the binary cross-entropy (BCE) loss, $\hat{L}_I$ is the radiance value at the second intersection point $\hat{x}$ obtained by querying NeuS, and $\hat{V}(\mathbf{x}, \boldsymbol{\omega})$ is obtained using an octree tracer from point $\mathbf{x}$ along direction $\boldsymbol{\omega}$.

## 3.3 Stage 2: BRDF Estimation

So far, we have faithfully reconstructed the prior information of the scene such as the normal, visibility and the indirect illumination. In this stage, we aim to accurately evaluate the rendering equation in

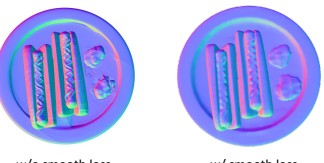
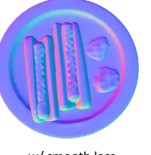

w/o smooth loss    w/ smooth loss

Figure 2: Smooth loss to fix broken part.

Input    Before Reg-Estim    After Reg-Estim

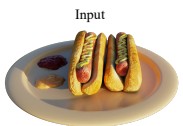 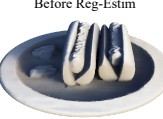 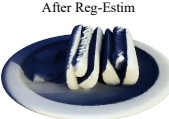

Figure 3: Visualization of direct SGs.

order to precisely estimate the surface BRDF i.e. albedo $a$, roughness $r$ and direct environment light with the fixed priors from stage 1. However, previous approaches tend to leave shadow and specular reflection in PBR materials under scenes with high illumination. Thus, we apply a scene-specific ACES tone mapping to the PBR color output by the rendering equation. The ACES tone mapping can calculate the PBR color over a broader value range, better estimating BRDF without baking shadow through more refined contrast control. We adopt SGs to efficiently approximate the rendering equation as PhySG [54]. See complete SGs approximation in the supplementary materials.

**Scene-specific ACES tone mapping.** We adopt the widely used the ACES tone mapping [1], which is a type of high dynamic range (HDR) tone mapping. Several recent works [16, 31] have incorporated HDR tone mapping into NeRF for specific applications. Specifically, we apply the ACES tone mapping $\mathcal{F}$ to convert the PBR color $e$ lying in $[0, +\infty)$ to color lying in $[0, 1]$:

$$\mathcal{F}(e) = \frac{(2.51e + 0.03)e}{(2.43e + 0.59)e + 0.14}, \tag{6}$$

whereas the ACES inverse tone mapping $\mathcal{F}_I$ is given by:

$$\mathcal{F}_I(c) = \frac{0.59c - 0.03 + \sqrt{-1.0127c^2 + 1.3702c + 0.0009}}{2(2.51 - 2.43c)}. \tag{7}$$

Given that the light intensity varies across different scenes, applying ACES tone mapping universally is not feasible. Thus, we introduce an additional learnable parameter $\gamma \in (0, 1]$. This scaled parameter modifies the ACES tone mapping curve, enabling it to automatically adapt to each scene's unique illumination intensity. The resulting deformed tone mapping function is defined as follows:

$$\mathcal{F}^\gamma(e) = \gamma^{-0.2}\mathcal{F}(e), \mathcal{F}_I^\gamma(c) = \mathcal{F}_I(c \cdot \gamma^{0.2}). \tag{8}$$

**Indirect illumination with scaled parameter.** In Sec. 3.2, we model the indirect illumination under the supervision from NeuS's radiance field. To convert indirect illumination to the same value range as BRDF estimation, we need to map the supervised values from NeuS through ACES inverse tone mapping $\mathcal{F}_I^\gamma$. Since we are not certain of the $\gamma$ that best fits the scene during stage 1, we train indirect illumination using randomly sampled $\gamma$ to obtain indirect illumination under all possible $\gamma$ settings. Consequently, the loss function $\mathcal{L}_{indir}$ in Eq. (5) is then revised to include $\gamma$ as follows:

$$\mathcal{L}_{indir} = \|\mathcal{F}_I^\gamma(\hat{L}_I) - L_I\|_1. \tag{9}$$

Then in stage 2, we stop training the indirect illumination and treat $\gamma$ as a learnable parameter. The optimal $\gamma$ for the current scene will be determined as the decomposition model converges.

**BRDF estimation.** We use the simplified Disney BRDF [6] model with albedo, roughness, and environment light as parameters and assume dielectric materials with a fixed Fresnel term value of $F_0 = 0.02$. During the BRDF estimation stage, we adopt $N = 128$ learnable SGs to model direct illumination and represent the PBR materials using an encoder-decoder network. The network initially encodes the input surface point $\mathbf{x}$ into its corresponding latent code $\mathbf{z}$ and then decodes it into albedo $\mathbf{a}$ and roughness $\mathbf{r}$. To further reduce noise in materials, we incorporate the smooth loss similar to Eq. (3) to both the albedo and roughness, and apply sparsity loss to $\mathbf{z}$ to ensure that most of the channels are close to zero:

$$\mathcal{L}_{smooth} = \|\mathbb{D}(\mathbf{z}), \mathbb{D}(\mathbf{z} + \epsilon)\|_2^2, \mathcal{L}_{sparse} = \text{KL}(\mathbf{z} \parallel 0.05), \tag{10}$$

where $\mathbb{D}$ is the decoder of the PBR material network, $\text{KL}(\rho \parallel \hat{\rho}) = \rho log\frac{\rho}{\hat{\rho}} + (1 - \rho)log\frac{1-\rho}{1-\hat{\rho}}$ represents Kullback-Leibler (KL) divergence loss that measures the relative entropy of two distributions.

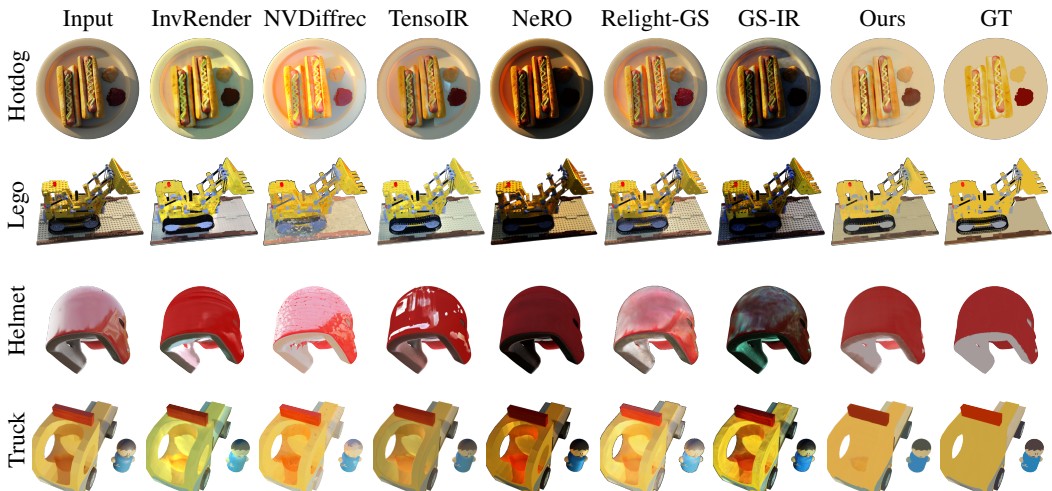

Figure 4: **Albedo in synthetic scenes.** We compare our method to InvRender [56], NVDiffrec [34], TensoIR [17], NeRO [26], Relightable-GS [11], and GS-IR [24]. The results show that our method outperforms previous approaches without baking specular highlights and shadows into albedo.

**SGs approximation for rendering equation.** In RobIR, we follow PhySG [54] and adopt SGs to approximate the rendering equation in Eq. (1):

$$
\begin{aligned}
f_r\left(\boldsymbol{\omega}_o, \boldsymbol{\omega}_i, \mathbf{x}\right) &= \frac{\mathbf{a}}{\pi} + f_s\left(\boldsymbol{\omega}_o, \boldsymbol{\omega}_i, \mathbf{r}\right) \\
\boldsymbol{\omega}_i \cdot \mathbf{n} &\approx G\left(\boldsymbol{\omega}_i; 0.0315, \mathbf{n}, 32.7080\right) - 31.7003, \\
L\left(\mathbf{x}, \boldsymbol{\omega}_i\right) &= \sum_{k=1}^{N} G\left(\boldsymbol{\omega}_i; \boldsymbol{\xi}_k, \lambda_k, \eta(\mathbf{x})\boldsymbol{\mu}_k\right) + \sum_{j=1}^{M} G_I\left(\boldsymbol{\omega}_i; \Gamma(\mathbf{x}, \gamma)\right),
\end{aligned}
\tag{11}
$$

where $G$ is the direct SGs learned in this stage, $G_I$ is the indirect SGs learned in stage 1, $\mathbf{n}$ is the surface normal, $\eta(\mathbf{x}) = \frac{\sum_{i=0}^{S} G(\boldsymbol{\omega}_i) V(\mathbf{x}, \boldsymbol{\omega}_i)}{\sum_{i=0}^{S} G(\boldsymbol{\omega}_i)}$ signifies the visibility for direct SGs obtained by randomly sampling $S$ directions, $f_s$ denotes the specular component that can be converted to a single SG. Then, we can integrate the multiplication of these SGs in closed-form [29] to compute the final PBR color $\boldsymbol{\omega}_o$. For more details about $f_s$, please see the supplementary materials.

### 3.4 Regularized Visibility Estimation

One of our primary goals is to achieve clean albedo with no residual shadows, which are typically caused by inaccurate visibility. Despite all efforts of the previous modeling, a small amount of stubborn visibility errors still exist. Therefore, after 100 epochs of BRDF estimation, we introduce regularized visibility estimation, directly using an MLP $\tilde{Q}(\mathbf{x}, \boldsymbol{\tau})$ to predict the visibility of $\mathbf{x}$ relative to $N$ direct SGs instead of $\eta$ calculated through previously learned visibility network $V(\mathbf{x}, \boldsymbol{\omega})$. Specifically, $\tilde{Q}(\mathbf{x}, \boldsymbol{\tau})$ is a visibility prediction network learned from scratch under the supervision of $\eta$, while $\boldsymbol{\tau}$ represents the $N \times N$ identity matrix used to add information for N direct SGs and $\mathbf{x} \in R^3$ is expanded to $R^{N \times 3}$ to predict visibility for each direct SG. Since visibility errors primarily occur at the edges, which are also sparse in the scene, we leverage the edge loss to make the residual sparse:

$$
\mathcal{L}_{edge} = \mathrm{KL}(\tilde{Q}(\mathbf{x}, \boldsymbol{\tau}) - \eta(\mathbf{x}) \,\|\, 0.01).
\tag{12}
$$

In the first 100 epochs, we fix $V(\mathbf{x}, \boldsymbol{\omega})$ using $\eta$ to obtain a stable visibility estimate, avoiding the early collapse of BRDF estimation caused by directly using $\tilde{Q}(\mathbf{x}, \boldsymbol{\tau})$. After 100 epochs, with a rough BRDF estimation in place, we introduce regularized visibility estimation. By using $V(\mathbf{x}, \boldsymbol{\omega})$ to distill $\tilde{Q}(\mathbf{x}, \boldsymbol{\tau})$, we directly predict the visibility of point $\mathbf{x}$ relative to direct SGs, circumventing errors caused by the sampling direction when calculating $\eta$. Thus, we can achieve a more accurate visibility estimate designed for direct SGs (See Fig. 3).

| | Albedo | | | Env Map | | | Relighting | | | Roughness |
|---|---|---|---|---|---|---|---|---|---|---|
| Method | PSNR ↑ | SSIM ↑ | LPIPS ↓ | PSNR ↑ | SSIM ↑ | MAE ↓ | PSNR ↑ | SSIM ↑ | LPIPS ↓ | MAE ↓ |
| NVDiffrec | 16.89 | 0.8252 | 0.1965 | 6.63 | 0.1397 | 0.3897 | 17.33 | 0.8235 | 0.2008 | 0.112 |
| InvRender | 19.12 | 0.8757 | 0.1652 | 13.47 | 0.5796 | 0.1624 | 22.57 | 0.8967 | 0.1354 | 0.073 |
| TensoIR | 20.52 | 0.8679 | 0.1537 | 5.19 | 0.4064 | 0.4903 | 18.66 | 0.8260 | 0.1981 | 0.066 |
| Relightable-GS | 17.63 | 0.8343 | 0.1695 | 9.96 | 0.3354 | 0.2413 | - | - | - | 0.104 |
| GS-IR | 14.88 | 0.7618 | 0.2170 | 5.10 | 0.1569 | 0.4530 | 17.18 | 0.8307 | 0.1891 | 0.142 |
| Ours-no aces | 21.24 | 0.8851 | 0.1421 | 10.50 | 0.5446 | 0.2379 | 23.61 | 0.9059 | 0.1221 | 0.065 |
| Ours-no rve | 18.51 | 0.8786 | 0.1403 | 10.10 | 0.5650 | 0.2486 | 23.20 | 0.8981 | 0.1243 | 0.059 |
| Ours-Log | 21.13 | 0.8883 | 0.1294 | 17.07 | 0.6431 | 0.1091 | 24.07 | 0.9003 | 0.1095 | 0.077 |
| Ours | 25.09 | 0.9303 | 0.0972 | 16.32 | 0.6351 | 0.1215 | 24.65 | 0.9118 | 0.0972 | 0.045 |

Table 1: **Quantitative evaluations.** We present the results of the synthetic scenes. We color each cell as best , second best , and third best . Our method can produce high-quality albedo, roughness, and environment map while maintaining the relighting fidelity.

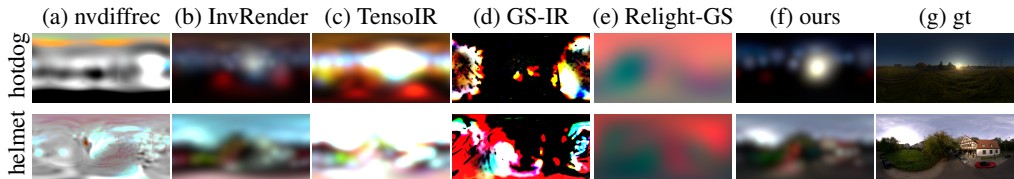

Figure 5: **Environment map.** Compared to existing approaches, our method can truly achieve high-quality environment light decoupling, avoiding messy results.

**Final loss.**    After incorporating regularized visibility estimation into inverse rendering, our final loss function in the BRDF estimation stage is:

$$\mathcal{L} = \|\mathcal{F}^\gamma(C_{pbr}), C_{gt}\|_2^2 + \lambda_{smooth}\mathcal{L}_{smooth} + \lambda_{sparse}\mathcal{L}_{sparse} + \lambda_{edge}\mathcal{L}_{edge}, \qquad (13)$$

where $C_{pbr}$ is the physically-based color from the rendering equation, $\mathcal{F}^\gamma$ is the scene-specific ACES tone mapping, $C_{gt}$ is the ground-truth color. In our experiments, $\lambda_{smooth}$, $\lambda_{sparse}$, and $\lambda_{edge}$ are set to 0.001, 0.01, and 1.0 respectively.

# 4 Experiments

In this section, we present the experimental evaluation of our methods. To assess the effectiveness of our approach, we collect synthetic and real-world datasets from NeRF and NeuS **without any post-processing**. In addition, we use Blender to render our own datasets to further demonstrate the superiority of our methods in high-illumination scenes. It should be noted that unlike previous methods [17, 55] that used a hotdog scene with reduced illumination, we use the original hotdog from NeRF [32] without reduced illumination. See more comparison in the supplementary materials.

Our model hyperparameters consisted of a batch size of 1024, with 200k iterations for the NeuS training. The model was implemented in PyTorch and optimized with the Adam optimizer at a learning rate of $5e^{-4}$. All tests were conducted on a single Tesla V100 GPU with 32GB memory. The training time without NeuS is around 5 hours.

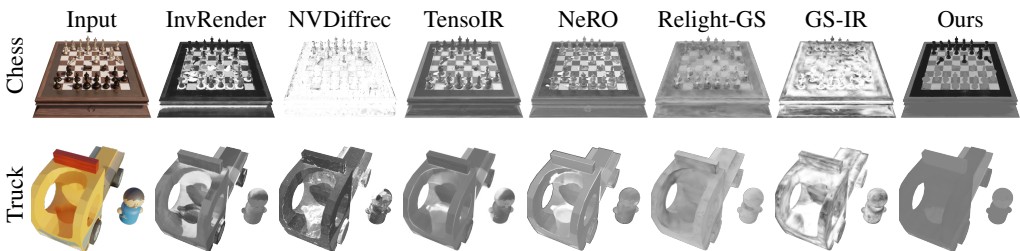

Figure 6: **Roughness in synthetic scenes.** The results show that our method can achieve clean roughness, even in scenes with intense shadow interference.

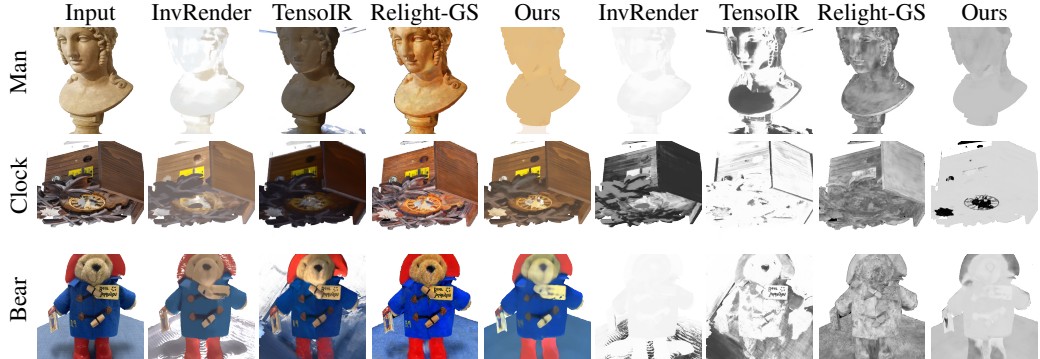

Figure 7: **Comparisons on real-world scenes.** Columns 2 to 5 are albedo, the last four columns are roughness. Even in complex real-world scenarios, our method can robustly decouple shadow and material, resulting in high-quality albedo and roughness.

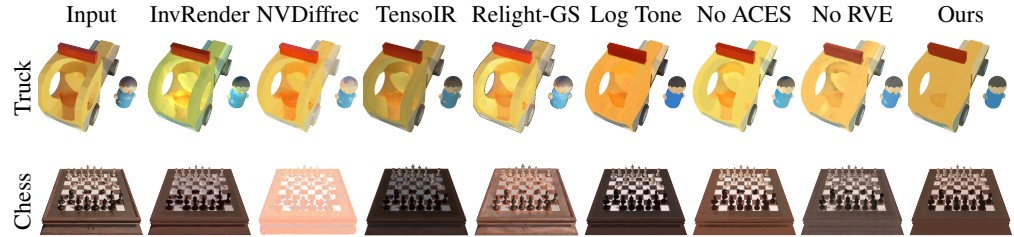

Figure 8: **Ablation.** We conduct ablation experiments on the key components in the BRDF estimation stage. The ablation results emphasize the critical importance of each component in our proposed framework for attaining high-quality albedo.

## 4.1 Comparisons with previous methods

We compare our method with previous state-of-the-art neural field-based inverse rendering approaches: NVDiffrec [34], InvRender [56], TensoIR [17], NeRO [26], Relightable-GS [11], and GS-IR [24].

As shown in Fig. 4 and Fig. 6, our method can truly achieve robust BRDF estimation, correctly decoupling shadows, ambient lighting, and PBR materials without baking shadows and specular highlights into albedo and roughness. Other methods tend to bake shadows into albedo, which also affects the correct decomposition of object roughness, reflecting their inability to properly separate the various components of BRDF estimation. Even in more challenging real-world scenarios shown in Fig. 7, our method can achieve robust decomposition results without baking shadows and specular highlights into albedo and roughness.

The estimated environment maps are shown in Fig. 5. Our method can accurately estimate the position of the light source and generate more precise light intensity in high-illumination scenes. As far as we know, we are the first to incorporate the accuracy of the estimated environment map into the quality assessment of neural field-based inverse rendering.

Tab. 1 shows the accuracy of the albedo, roughness, relighting, and environment map averaged over synthetic scenes. We did not measure the relighting of Relightable-GS because it does not support relighting of a single object. The term "Log" refers to the use of sigmoid mapping instead of ACES. We can observe that our method achieve the best results in all inverse rendering tasks. Inaccurate BRDF estimation significantly affects the results of relighting, causing methods with high-quality reconstruction to bake shadows and thus leading to a decline in rendering quality during relighting. Overall, our approach can achieve robust inverse rendering in high-illumination scenes.

## 4.2 Ablation Studies

We perform an ablation study to analyze the importance of the key components in our proposed method. As illustrated in Fig. 8, our method is unable to eliminate both shadows or specular

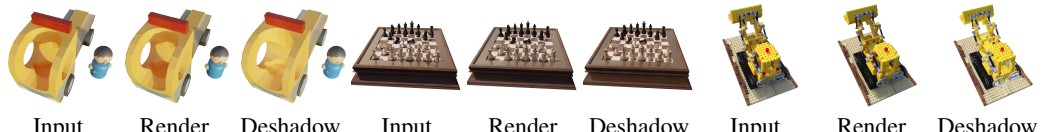

| Input | Render | Deshadow | Input | Render | Deshadow | Input | Render | Deshadow |

Figure 9: **De-shadow.** Given an input image from a specific viewpoint, our proposed method can accurately remove shadows caused by direct light occlusion without sacrificing rendering quality.

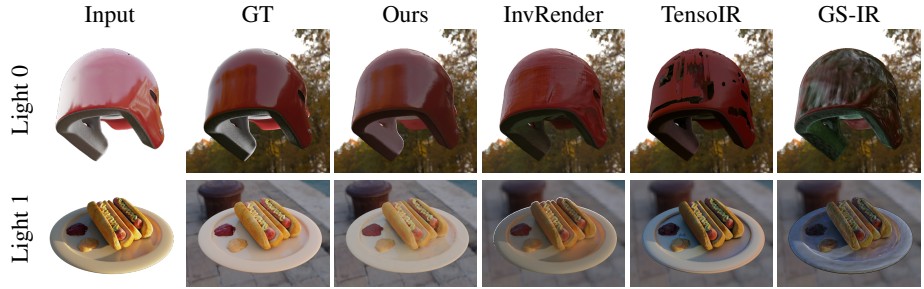

Figure 10: **Relighting.** Our method not only achieves high-quality relighting results in scenarios with specular highlights but can also robustly decouple shadows, obtaining high-quality relighting outcomes without baked shadows even in scenes with severe shadows.

reflection in the absence of ACES tone mapping. Without regularized visibility estimation, inaccurate predictions of direct SGs visibility results in residual shadows. The "Log Tone" result indicates that ACES is a more effective tone mapping than the sigmoid to remove shadow within our framework. Finally, our full method can correctly estimate BRDF of the object, resulting in the best performance.

### 4.3 Application

**De-shadowing.** De-shadowing is a challenging task in the field of inverse rendering, often requiring strong priors and large data-driven models. Our proposed method correctly understands various lighting effects and is capable of effectively eliminating strong and irregular shadows, particularly in scenes with intense lighting. As shown in Fig. 9, by setting the visibility of direct SGs to 1, we can remove the shadow caused by direct light occlusion. It should be noted that our method **cannot remove the areas with reflections and the dark regions caused by the backlighting phenomenon**.

**Relighting.** To demonstrate the practical utility of the materials from our method, we conducted relighting experiments. As shown in Fig. 10, our estimated BRDF results can be accurately relighted in various lighting environments without shadow or illumination artifacts.

## 5 Conclusions and Discussions

We presented a novel inverse rendering framework for estimating BRDF of the object under high-illumination scenes. The key innovation lies in the use of ACES tone mapping, which shifts the calculation of PBR color to a wider value range, significantly reducing the impact of shadows and specular parts on BRDF estimation. In addition, regularized visibility estimation are employed to ensure more acuurate visibility for direct SGs. Experiment results on both synthetic and real-world data show that our method outperforms previous approaches in eliminating shadows and specular reflection under high-illumination scenes.

Currently, the proposed method has some limitations. First, non-solid, translucent, and thin objects cannot be correctly handled due to the limitations of NeuS. Second, the employment of SGs to model both direct and indirect lighting presents challenges in dealing with anisotropic objects, consequently leading to our method's deficiency in incorporating the metallic learnable parameters present in the Disney BRDF model. Third, we have not considered scenes with dynamic lighting like [28, 44]. Finally, our method's prior information is limited to multi-view images. We will consider integrating with LLM models in the future work.

# 6 Acknowlegements

This work was supported by Key R&D Program of Zhejiang (No. 2024C01069). We thank Wenxin Sun for her help in pipeline illustration. We also thank Yuan Liu and Wen Zhou for the constructive suggestions.

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

## Appendix

This supplementary document provides some implementation details and further results that accompany the paper.

- Section A introduces the differences between the dataset used by our method and those used by previous methods.

- Section B introduces more details of the SG approximation for the rendering equation.

- Section C provides additional results, including more visualizations and results on more datasets.

## A  High-illumination Dataset

Currently, neural field-based inverse rendering methods, such as InvRender [56], NeRFactor [55], and TensoIR [17], generally use scenes with almost no high-intensity ambient light (See Fig. 13). The advantage of these scenes is that the object's BRDF estimation is not affected by self-occlusion shadows, making albedo and color quite similar. As a result, even if each part of the BRDF estimation is somewhat messy, plausible results can still be obtained. However, when the scene has intense illumination and shadows, these methods will fail to correctly model the object's BRDF. Therefore, to more accurately evaluate the robustness of inverse rendering, we choose a more challenging high-illumination dataset.

## B  SG Approximation for the Rendering Equation

Following the methodology from [54], we employ the inner product of SGs to approximate the computation of the rendering equation. The position $\mathbf{x}$ is dropped in the following equation due to the distant illumination assumption. Specifically, the term $\boldsymbol{\omega}_i \cdot \mathbf{n}$ is approximated by a SG as follows:

$$\boldsymbol{\omega}_i \cdot \mathbf{n} \approx G\left(\boldsymbol{\omega}_i; 0.0315, \mathbf{n}, 32.7080\right) - 31.7003. \tag{14}$$

As for the specular component $f_s$, we employ the simplified Disney BRDF model as previous methods [6, 23, 2]:

$$f_s\left(\boldsymbol{\omega}_o, \boldsymbol{\omega}_i\right) = \mathcal{M}\left(\boldsymbol{\omega}_o, \boldsymbol{\omega}_i\right) \mathcal{D}(\mathbf{h}),$$
$$\mathbf{h} = \frac{\boldsymbol{\omega}_o + \boldsymbol{\omega}_i}{\|\boldsymbol{\omega}_o + \boldsymbol{\omega}_i\|_2}, \tag{15}$$

where $\mathcal{M}$ represents the Fresnel with shadowing effects, and $\mathcal{D}$ is the normalized distribution function.

To simplify the computation, we assume an isotropic specular BRDF, and adapt $\mathcal{D}$ and $\mathcal{M}$ as follows:

$$\mathcal{M}\left(\boldsymbol{\omega}_o, \boldsymbol{\omega}_i\right) = \frac{\mathcal{F}\left(\boldsymbol{\omega}_o, \boldsymbol{\omega}_i\right) \mathcal{G}\left(\boldsymbol{\omega}_o, \boldsymbol{\omega}_i\right)}{4\left(\mathbf{n} \cdot \boldsymbol{\omega}_o\right)\left(\mathbf{n} \cdot \boldsymbol{\omega}_i\right)}$$
$$\mathcal{F}\left(\boldsymbol{\omega}_o, \boldsymbol{\omega}_i\right) = \boldsymbol{s} + (1 - \boldsymbol{s}) \cdot 2^{-(5.55473\boldsymbol{\omega}_o \cdot \mathbf{h} + 6.8316)(\boldsymbol{\omega}_o \cdot \mathbf{h})},$$
$$\mathcal{G}\left(\boldsymbol{\omega}_o, \boldsymbol{\omega}_i\right) = \frac{\boldsymbol{\omega}_o \cdot \mathbf{n}}{\boldsymbol{\omega}_o \cdot \mathbf{n}(1 - k) + k} \cdot \frac{\boldsymbol{\omega}_i \cdot \mathbf{n}}{\boldsymbol{\omega}_i \cdot \mathbf{n}(1 - k) + k},$$
$$k = \frac{(r + 1)^2}{8},$$
$$\mathcal{D}(\mathbf{h}) = G\left(\mathbf{h}; \mathbf{n}, \frac{2}{r^4}, \frac{1}{\pi r^4}\right),$$

where $s \in [0, 1]^3$ is the specular factor, and $r$ denotes the roughness. Finally, we can compute the rendering equation through the fast inner product of SGs [29].

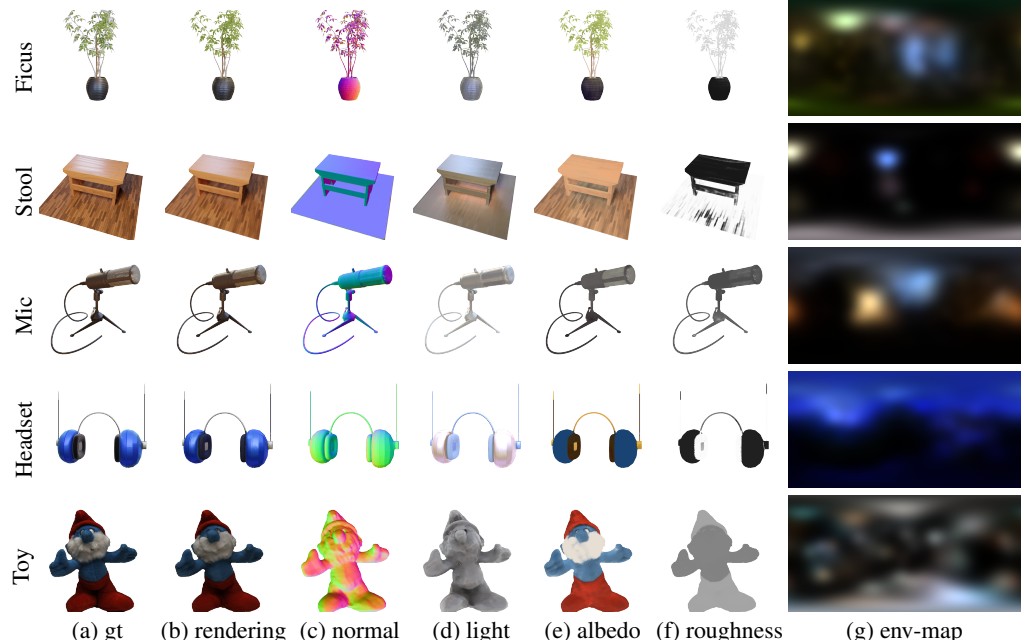

Figure 11: **Other results of our method.** In each scene, we present the input ground-truth image (a), our rendering result (b), normal (c), light (d), albedo (e), and roughness (f) obtained through our method. These experiments illustrate the generalizability of our method across diverse datasets and demonstrate its ability to produce high-quality results.

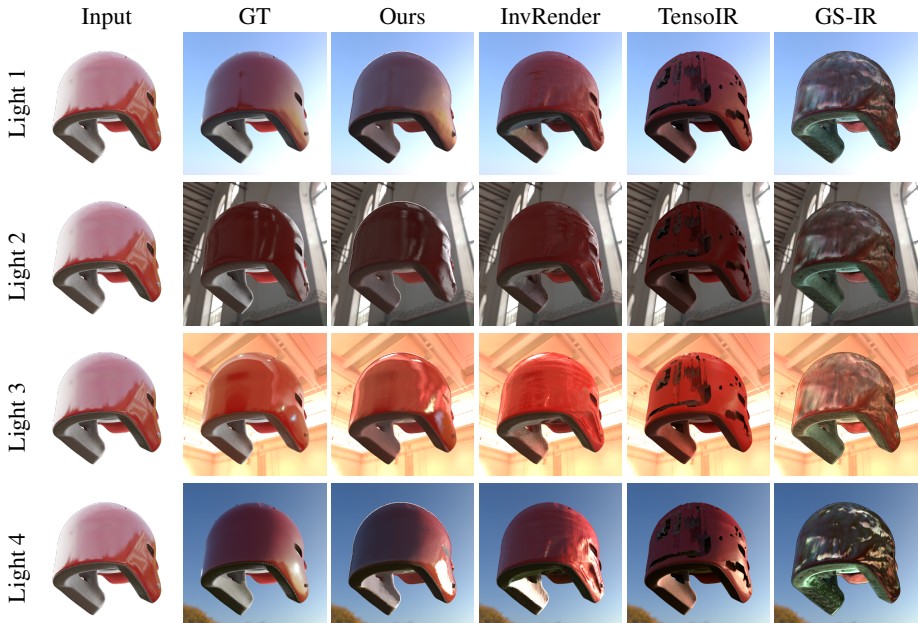

Figure 12: **Helmet Relighting.** Our method achieves high-quality relighting results in scenarios with specular highlights and slight specular reflections.

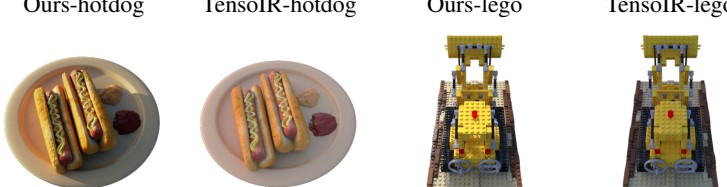

Ours-hotdog     TensoIR-hotdog     Ours-lego     TensoIR-lego

Figure 13: **Dataset Comparison.** We choose a more challenging high-illumination dataset, which exposed the inability of previous neural field-based inverse rendering methods to decouple shadows from the object's PBR materials.

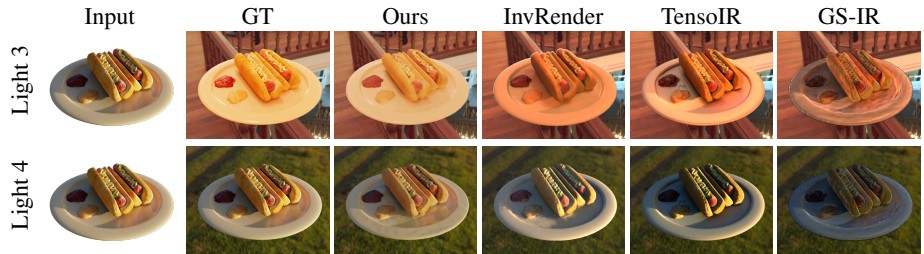

Figure 14: **Hotdog Relighting.** Our method achieves high-quality relighting results in scenarios with severe shadows.

| | Hotdog | | | Lego | | | Helmet | | |
|---|---|---|---|---|---|---|---|---|---|
| Method | PSNR↑ | SSIM↑ | LPIPS↓ | PSNR↑ | SSIM↑ | LPIPS↓ | PSNR↑ | SSIM↑ | LPIPS↓ |
| NVDiffrec | 20.60 | 0.8872 | 0.1777 | 18.52 | 0.8299 | 0.1616 | 12.06 | 0.7866 | 0.2274 |
| InvRender | 15.76 | 0.8575 | 0.2029 | 20.75 | 0.8606 | 0.1656 | 19.50 | 0.8761 | 0.1697 |
| TensoIR | 16.01 | 0.8496 | 0.2047 | 20.74 | 0.8493 | 0.1541 | 16.95 | 0.8341 | 0.1759 |
| Relightable-GS | 15.34 | 0.8453 | 0.2111 | 20.07 | 0.8030 | 0.1580 | 14.97 | 0.7946 | 0.1963 |
| GS-IR | 9.72 | 0.6382 | 0.3139 | 13.03 | 0.6860 | 0.2386 | 13.72 | 0.7774 | 0.2538 |
| Ours-no aces | 22.24 | 0.8582 | 0.1779 | 22.00 | 0.8675 | 0.1497 | 19.98 | 0.9173 | 0.1202 |
| Ours-no rve | 19.04 | 0.8487 | 0.1489 | 20.17 | 0.8659 | 0.1437 | 14.30 | 0.8769 | 0.1493 |
| Ours-Log | 19.01 | 0.8570 | 0.1560 | 21.43 | 0.8596 | 0.1504 | 21.87 | 0.9078 | 0.1012 |
| Ours | 24.25 | 0.9185 | 0.0970 | 24.63 | 0.9175 | 0.1071 | 24.14 | 0.9427 | 0.1122 |
| | Truck | | | Stool | | | Average | | |
| Method | PSNR↑ | SSIM↑ | LPIPS↓ | PSNR↑ | SSIM↑ | LPIPS↓ | PSNR↑ | SSIM↑ | LPIPS↓ |
| NVDiffrec | 19.59 | 0.9010 | 0.1437 | 13.69 | 0.7213 | 0.2722 | 16.89 | 0.8252 | 0.1965 |
| InvRender | 21.68 | 0.9023 | 0.1440 | 17.90 | 0.8822 | 0.1440 | 19.12 | 0.8757 | 0.1652 |
| TensoIR | 24.06 | 0.9375 | 0.1071 | 24.81 | 0.8692 | 0.1269 | 20.51 | 0.8679 | 0.1537 |
| Relightable-GS | 19.17 | 0.8720 | 0.1528 | 18.62 | 0.8567 | 0.1296 | 17.63 | 0.8343 | 0.1696 |
| GS-IR | 19.03 | 0.8379 | 0.1729 | 18.92 | 0.8695 | 0.1061 | 14.88 | 0.7618 | 0.2171 |
| Ours-no aces | 20.81 | 0.9177 | 0.1111 | 21.15 | 0.8647 | 0.1518 | 21.24 | 0.8851 | 0.1421 |
| Ours-no rve | 20.59 | 0.9200 | 0.1108 | 18.46 | 0.8814 | 0.1511 | 18.51 | 0.8786 | 0.1408 |
| Ours-Log | 24.04 | 0.9418 | 0.0997 | 19.29 | 0.8755 | 0.1395 | 21.13 | 0.8883 | 0.1294 |
| Ours | 27.46 | 0.9592 | 0.0647 | 24.98 | 0.9136 | 0.1051 | 25.09 | 0.9303 | 0.0972 |

Table 2: **Quantitative albedo comparison on synthetic dataset**. We compare our method to several previous approaches: NVDiffrec [14], InvRender [56], TensoIR [17], Relightable-GS [11] and GS-IR [24]. We report PSNR, SSIM, LPIPS(VGG) and color each cell as best , second best and third best .

## C    Additional Results

**More qualitative results.** Our method can effectively remove shadows baked into albedo and roughness, thanks to our accurate modeling of each decomposition component. Therefore, our method can certainly handle scenes with less intense lighting. Fig. 11 shows the results of our method on real-world datasets and some synthetic datasets, including scenes with shadows and specular, as

well as diffuse objects. Our method can robustly perform inverse rendering in any situation without baking shadows and illumination into PBR materials.

**Per-scene albedo results.** We present the complete metrics of our method compared to other methods in Tab. 2. The estimated albedo in our method surpasses existing SOTA methods in every synthetic scene.

**More relighting.** We show more relighting results in Fig. 12 and Fig. 14. The two scenes demonstrate that our method can accurately estimate the BRDF of the object under scenes with specular highlights and severe shadows.

**Additional comparison with NVDiffrecMC.** We show more albedo, roughness, and environment map comparison with NVDiffrecMC [14] in Figs. 15-17.

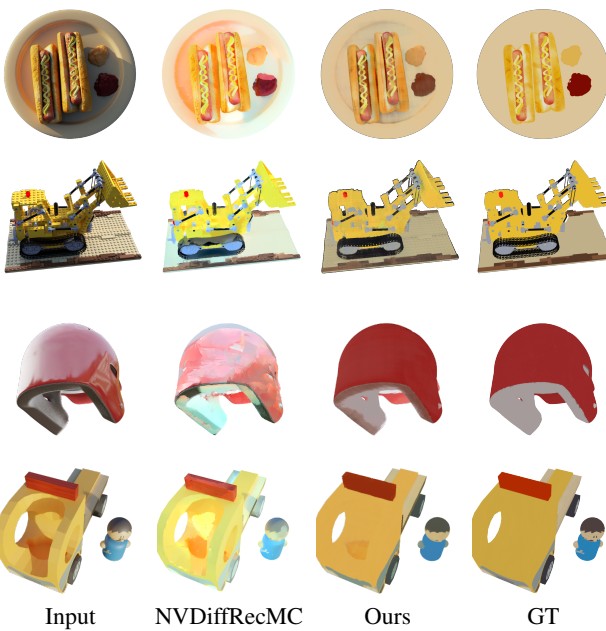

Input         NVDiffRecMC         Ours         GT

Figure 15: **Albedo comparison with NvDiffRecMC on synthetic scenes.** NvDiffRecMC cannot achieve the decouple of shadow, indirect illumination, and the PBR materials of the objects.

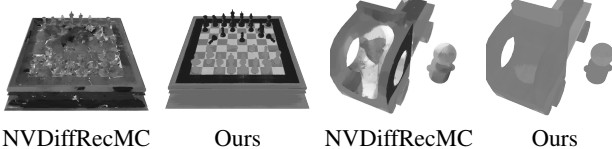

NVDiffRecMC         Ours         NVDiffRecMC         Ours

Figure 16: **Roughness comparison with NvDiffRecMC.** NvDiffRecMC cannot obtain high-quality roughness in high illumination scenes.

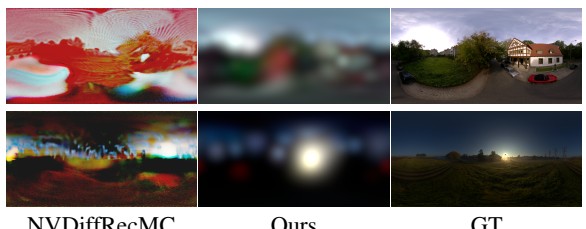

NVDiffRecMC         Ours         GT

Figure 17: **Environment map comparison with NvDiffRecMC.**

**Visualization and evaluation on tone mapping.** We first visualize the vanilla ACES and sRGB tone mapping in Fig. 18, indicating that ACES curve has much wider input range. Then in Fig. 19, we show the scene-specific tone mapping curve with different $\gamma$, enabling the ACES curve to fit other settings with different tone mapping methods. Finally, we evaluate the optimized ACES curve (with $\gamma = 0.42$) in *chessboard* scene with GT tone mapping (sRGB) in Fig. 20. The results show that our scene-specific ACES tone mapping can stretch to sRGB curve, demostrating the effectiveness of our method.

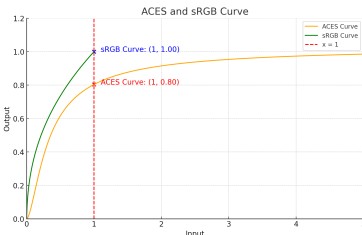

Figure 18: **Comparison on ACES and sRGB curve.**

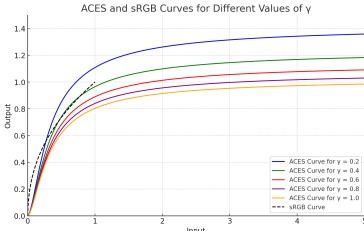

Figure 19: **Visualization of ACES tone mapping with different $\gamma$.**

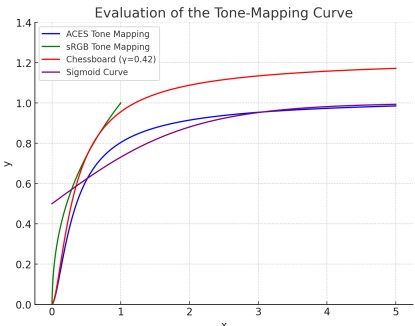

Figure 20: **Evaluation tone mapping in chessboard.** The ACES tone mapping with $\gamma = 0.42$ matches well with the sRGB curve.

