# OpenReview forum: "RobIR: Robust Inverse Rendering for High-Illumination Scenes"
_NeurIPS.cc/2024/Conference — NeurIPS 2024 poster_

### Official Review · Reviewer_ijRb · 2024-07-11

**Soundness:** 3
**Presentation:** 3
**Contribution:** 3
**Rating:** 7
**Confidence:** 4

**Summary:**

This paper addresses inverse rendering in high-illumination scenes with strong shadows where past methods bake shadows and highlights into estimation results. This paper proposes to use ACES tone mapping and makes it scene-dependent for inverse rendering in high-illumination scenes. This paper also proposes to directly estimate the visibility of each spherical Gaussian of direct illumination instead of a visibility field, which enables an accurate representation of shadows at the edge. The experimental results on the synthetic and real-world datasets show that the proposed method can estimate accurate albedos, surface roughness, and illumination without artifacts in the high-illumination scenes.

**Strengths:**

+ This paper proposes a novel regularized visibility estimation that enables an accurate representation of shadows at the edge.
+ Experimental results show that the proposed method successfully estimates BRDF and illumination while existing methods suffer from artifacts. They also indicate the effectiveness of ACES tone mapping compared with log tone mapping methods.

**Weaknesses:**

- It is unclear why the ACES tone mapping, of which usage is the key contribution, enables the robust inverse rendering of high-illumination scenes with strong shadows.
- The proposed method loses the albedo of the detailed texture due to smoothness loss in Eq. 10. For example, Bear in Fig. 7 and truck in Fig. 4.

**Questions:**

- A more detailed explanation of the effects of ACES tone mapping is expected. Why is it more suitable for high-illumination scenes than other tone mapping methods such as sRGB and log tone mapping? How does it affect the loss and the optimization?

**Limitations:**

Yes.

---

> ### Author Rebuttal · Authors · 2024-08-05
>
> We are glad and appreciate that the reviewer recognizes the novel ideas of RobIR and the proposed method successfully estimates BRDF and illumination. Since many of the questions have already been answered in the common response, our additional response to the reviewer’s comments is below:
>
> **Q1: Why ACES is more suitable than other tone mapping like sRGB and sigmoid.**
>
> As previously mentioned in the common response, high-illumination scenes often require HDR tone mapping to produce the final image, mapping values from 0 to A into the range of $[0, +\infty]$. However, sRGB is not designed for processing HDR input. ACES allows PBR output colors to be in a larger value range, which better distinguishes very dark shadow regions and aids in the decoupling of object materials.
>
> Additionally, ACES has significant advantages over sigmoid curves. Although both map an infinite range to $[0,1]$, ACES is a more commonly used HDR tone mapping curve in graphics, consistent with many rendering engines, and provides better contrast control while preserving more detail in highlights and shadows.
>
> **Q2: Lost details in albedo.**
>
> As previously mentioned in the common response, although the smooth loss does cause some loss of detail, the rendering precision of NeuS is a bigger factor in this loss. In the submitted rebuttal PDF, we included the results of NeuS rendering, which clearly show some blurring. This blurring affects other BRDF components supervised by NeuS, exacerbating the over-smoothing issue. Therefore, we are very eager to use methods with higher rendering and geometric quality, such as 2DGS, in future work.

---

> ### Author Response · Authors · 2024-08-08
> **Fix typo errors in Q1**
>
> **Q1: Why ACES is more suitable than other tone mapping like sRGB and sigmoid.**
>
> As previously mentioned, high-illumination scenes often require HDR tone mapping to produce the final image, mapping values from $[0, +\infty]$ into the range of $[0, 1]$. However, sRGB is not designed for processing HDR input. ACES allows PBR output colors to be in a larger value range, which better distinguishes very dark shadow regions and aids in the decoupling of object materials.
>
> Additionally, ACES has significant advantages over sigmoid curves. Although both map an infinite range to $[0,1]$, ACES is a more commonly used HDR tone mapping curve in graphics, consistent with many rendering engines, and provides better contrast control while preserving more detail in highlights and shadows.

---

> > ### Comment · Reviewer_ijRb · 2024-08-11
> > **Additional questions**
> >
> > I appreciate your addressing the questions. After reading the rebuttal, I have additional questions about tone mapping to clarify the contributions of this paper.
> > 1. Does the proposed method assume an unknown method of tone mapping for the input images? If so,
> > * Is this assumption common for applications of inverse rendering?
> > * Is the proposed scene-dependent tone mapping not needed when the tone mapping method is known and can be used in optimization?
> > 2. Would you like to claim that the existing inverse rendering method misses the mismatch of tone mapping between image processing (image formation) and optimization, resulting in failure in high-illumination scenes?

---

> ### Author Response · Authors · 2024-08-11
>
> Thank you very much for the constructive feedback. We hope that our response below will address your concerns.
>
> **Q1: Unknown tone mapping assumption.**
>
> Our proposed method only takes a collection of images with camera poses as input, and the ground truth tone mapping of these images is unknown. All the methods we compare in our paper use datasets with unknown ground truth tone mapping. We introduce scene-specific ACES tone mapping to approximate various tone mapping scenarios; for example, HDR scenes are typically optimized to vanilla ACES tone mapping. Therefore, if the tone mapping method were known, theoretically, we wouldn’t need to use our proposed scene-specific ACES. However, this is nearly impossible because different physics engines employ different tone mappings, and tone mapping itself involves numerous parameters. Additionally, variations camera parameters in real-world scenes make it even more difficult for us to determine the ground truth tone mapping.
>
> **Q2: The problem of the existing inverse rendering method.**
>
> The existing inverse rendering methods, such as NeRO, use sRGB tone mapping. However, in high-illumination scenes, we typically need tone mapping that can handle HDR inputs to produce the final image. We are the **first** to apply ACES tone mapping in inverse rendering, which fundamentally enables our method to handle inverse rendering in high-illumination scenes. Additionally, the introduction of scene-specific ACES tone mapping allows us to approximate other tone mappings (e.g., sRGB), making our method compatible with non-high-illumination scenes as well.
>
> Building on more accurate tone mapping, we achieve more precise visibility modeling through Regularized Visibility Estimation (RVE) and NeuS Octree, which effectively removes shadows and disentangles other parts of the BRDF estimation more accurately.
>
>
> Therefore, the failure (especially shadow baking) of existing inverse rendering methods in high-illumination scenes stems from 1) the mismatch of tone mapping between image processing and optimization, as well as 2) inaccurate visibility estimation.

---

> > ### Comment · Reviewer_ijRb · 2024-08-12
> >
> > Thank you for your response. I acknowledge the novel idea of scene-specific tone mapping to handle both HDR and non-HDR scenes as well as the effectiveness of the visibility estimation. I raise my rating from Borderline reject to Accept.

---

> > > ### Author Response · Authors · 2024-08-13
> > >
> > > Thank you for your recognition and adjustment of the score. We are glad that our rebuttal effectively addressed your concerns. We also appreciate your recognition of our methods. Thank you again for your constructive feedback and guidance throughout the review process.

---

### Official Review · Reviewer_QVta · 2024-07-12

**Soundness:** 3
**Presentation:** 3
**Contribution:** 3
**Rating:** 5
**Confidence:** 5

**Summary:**

This paper proposes a method for the inverse rendering of high-illumination and highly reflective scenes. There are two training phases, in the first phase, it trains by Neus, to get geometry and compute visibility by octrees. In the second phase, it decomposes lighting as SGs and material by MLPs.

**Strengths:**

Direct and indirect lighting, visibilities are presented by SGs, which is compact.
From the results, shadows and specularities are decomposed well.
Tone mapping is used, as NeRF in the Dark, which improves results for high-illumination scenes.

**Weaknesses:**

Figure 2 gives comparisons with and without smooth loss, however, the one w/o smooth loss is better, while the loss may over-smooth the details.
Figure 11 shows results where the lighting consists of the original colors.

**Questions:**

How does the relighting work? Do you use original indirect lighting?
Why is visibility divided into two stages, and why not directly use the results calculated by the Neus octree as the ground truth to supervise SG? Instead, why is the Neus octree result used as the ground truth to supervise the MLP, and then the MLP's distribution used to supervise the SG results? What are the differences?

**Limitations:**

See above.

---

> ### Author Rebuttal · Authors · 2024-08-03
>
> We thank the reviewer for the positive review as well as the suggestions for improvement. Our response to the reviewer’s comments is below:
>
> **Q1: How does the relighting work.**
>
> We imported the reconstructed albedo and roughness maps into Blender and performed relighting using Blender's scripting. Fig. 10 aims to demonstrate that we can extract high-quality PBR materials that can be directly applied in creative tools like Blender.
>
> **Q2: Why use an MLP instead of directly using the results from the NeuS Octree.**
>
> Good question. This is because querying Octree tracing is extremely time-consuming. By using an MLP, we can cache the results of the NeuS Octree and speed up inference.
>
> **Q3: Over-smooth details in Fig. 2.**
>
> This is a finding we want to highlight: smoothing helps correct geometric errors. Although the smooth loss results in some loss of detail, it effectively fixes geometric breakages caused by reflections or shadows in the normal map. Addressing these issues will aid in the decoupling of shadows and indirect illumination.
>
> **Q4: Environment map consists of the object color.**
>
> The estimation of environment map is still not robust in current IR frameworks. However, we want to emphasize that, as shown in Figure 5, our method has significantly improved the accuracy of environment map estimation compared to previous approaches. Additionally, we are one of the few methods that quantitatively compare environment map metrics (see Table 1). We will focus on addressing the estimation of environment maps in cases like those shown in Figure 11 in our future research.

---

### Official Review · Reviewer_GRUR · 2024-07-13

**Soundness:** 3
**Presentation:** 3
**Contribution:** 3
**Rating:** 5
**Confidence:** 5

**Summary:**

This paper introduces RobIR, an inverse rendering approach that can better tackle “high-illumination” scenes. RobIR first leverages the existing neural field model (NeuS) to represent 3D geometry information including normal, visibility, and indirect illumination. It then utilizes these geometry priors to decompose rendering attributes of the scene through the approximated rendering equation with spherical Gaussian. This work introduces an optimizable ACES tone-mapping and regularized visibility estimation model to better handle HDR color and shadow occlusions, respectively. Their experiment demonstrates some impressive results on shadow disentanglement.

**Strengths:**

1. This work has some further thoughts on color tone mapping for the typical multi-view inverse rendering. The proposed optimizable ACES tone mapping looks very effective in improving inverse rendering results.
2. The proposed visibility representation (RVE) also plays an important role in the final results. RVE with its neural net accumulates and denoises Monte Carlo samples to achieve more accurate and stable visibility evaluation. Their efforts to refine the visibility should be appreciated.
3. I am glad the authors clearly point out they use the original NeRF rendering of Hotdog and Lego, instead of the NeRFactor’s.

**Weaknesses:**

1. This paper still follows a commonly used multi-stage inverse rendering strategy with neural fields. The geometry representation is based on NeuS; the rendering formulation (SG rendering, visibility, and indirect illumination) is mainly based on InvRender; The proposed optimizable ACES tone and REV are more like incremental improvements over InvRender. The key rendering formulation and optimization remain the same as the prior works. Therefore, the novelty of this work is moderate.
2. The description of RVE in Sec. 3.4 is not very clear.  It seems that MLP $Q(x, \tau)$ directly outputs N visibility ratios, thus $\eta(x)$ in Eq. 12 should also be an N-dim vector instead of a scalar value.
3. The proposed method is quite time-consuming. Training time even without NeuS is around 5 hours.
4. The proposed regularization terms may hurt the high-frequency details in real-world scenes (Fig. 7).
5. This method is limited to dielectric materials, without the consideration of metallic and glossy objects, as already pointed out by the authors.
6. The paper should include some inverse rendering methods with differentiable path tracing, as these methods can explicitly handle visibility, for example, NvDiffRecMC, Mitsuba, etc.
7. Minor errors:
    * L275 accuurate -> accurate
    * Figure 8: Hotdog label is wrong.

**Questions:**

1. What are the differences between the final optimized ACES tone mapping and sRGB gamma tone mapping? How does this optimized tone mapping vary from scene to scene? It would be better if tone-mapping curves were included in the paper.
2. I tested this ACES tone mapping mentioned in the paper, it seems that the proposed learnable ACES tonemapping (an S-curve) cannot well approximate the existing concave tone-mapping curves that are potentially used for rendering Blender objects (e.g., sRGB curve, Filmic curve, AgX curve, etc.). Given this limitation, how does the paper address the color mismatch in the low-illumination color space?
2. The paper does not show the metrics for rendering (NVS) with decomposed attributes. I’d like to see the rendering quality metrics for those NeRF scenes.
3. It would be better if the author could show some video examples of moving shadows while rotating envmaps in the final release.
4. Why does the learnable parameter $\gamma$ have an exponent 0.2?

**Limitations:**

The limitations are adequately discussed in the last paragraph of the paper.

---

> ### Author Rebuttal · Authors · 2024-08-05
>
> We thank the reviewer for the positive review as well as the suggestions for improvement. We will revise the typo errors in the paper based on these insightful suggestions. Our response to the reviewer’s comments is below:
>
> **Q1: Comparison with inverse rendering methods with differentiable path tracing.**
>
> Great suggestion. We will add citations to these papers in our paper. We have shown a comparison with `NvDiffRecMC` in the rebuttal PDF. As can be seen, NvDiffRecMC still cannot remove baked shadows and indirect illumination from PBR materials, demonstrating the robustness of our method.
>
> **Q2: Show some video examples of moving shadows while rotating envmaps.**
>
> Of course, no problem. Unfortunately, we are unable to submit a video during the rebuttal period. However, we can provide other examples to demonstrate that our results are indeed strong. In Fig. 9, we offer several examples that illustrate the robustness of our method in shadow removal, which can partially demonstrate our capability to achieve the task you mentioned.
>
> **Q3: This method is limited to dielectric materials.**
>
> We are very grateful to the reviewer for pointing out this issue and noticing that we discussed it in the Limitations section. We want to emphasize that our approach is fundamentally a general method that can be integrated with other NeuS-based inverse rendering methods for glossy objects, such as NeRO. In the NeRO paper, there is also an issue with shadow baking in PBR materials for glossy objects. We believe that our approach can help NeRO achieve higher quality PBR material decomposition for metallic and glossy objects.
>
> **Q4: Novel view synthesis on NeRF synthetic scenes.**
>
> From our perspective, we believe that novel view synthesis (NVS) is not the primary focus of inverse rendering. Instead, we are more concerned with relighting and the decoupled BRDF components and PBR materials. Using the rendering equation to constrain the process can negatively impact NVS performance, resulting in rendering metrics that are less competitive than methods that directly output color. Additionally, NVS performance is highly sensitive to the base method used; for example, approaches based on NeuS are likely to perform worse in NVS compared to methods based on TensoRF or 3D-GS, which can introduce a degree of unfairness. Despite this, we still provide NVS results (on hotdog, lego, ficus, and mic) to demonstrate that the fidelity of our reconstruction is acceptable.
>
> |           | PSNR  | SSIM   | LPIPS  |
> | --------- | ----- | ------ | ------ |
> | Ours      | 30.11 | 0.9466 | 0.0528 |
> | NVDiffrec | 29.81 | 0.9732 | 0.0345 |
> | InvRender | 27.64 | 0.9056 | 0.0982 |
> | TensoIR   | 32.47 | 0.9637 | 0.0377 |
>
> We want to emphasize once again that, although our NVS quality may not be the highest, the image supervision is sufficient to decouple high-quality and smooth PBR materials. These can be effectively used for tasks that are of greater interest in inverse rendering, such as relighting and shadow removal.
>
> **Q5: Why does the learnable parameter $\gamma$ have an exponent 0.2.**
>
> This is a very detailed question. It's an engineering trick to allow ACES to stretch across a wider range.
>
> **Q6: The use of $\eta (x)$ in Eq. 12 is inaccurate.**
>
> Thank you very much for pointing out this issue. Your understanding is completely correct. In Eq. 12, the supervision value for $Q(x, \tau)$ should be an N-dimensional vector, representing the visibility of N direct SGs at coordinate **x**. We will replace $\eta (x)$ with a different mathematical symbol in the paper.
>
>
>
> Finally, we would like to thank the reviewer once again. Many of the reviewer's comments were very accurate, which truly made us very happy.

---

> > ### Comment · Reviewer_GRUR · 2024-08-12
> >
> > Thanks for the authors' detailed responses. I am generally satisfied with these responses.
> > I'll retain or possibly increase my score later.
> > I still think NVS rendering quality is also an important factor that should be considered. I encourage you to include these metrics in the supplement.

---

> > > ### Author Response · Authors · 2024-08-13
> > >
> > > We thank the reviewer for the valuable comments and are keen to follow up on the provided suggestion to include NVS metrics in the supplement.
> > >
> > > Additionally, we want to emphasize that the NVS results are highly dependent on the base method. We tried applying our method to 3D-GS, and the NVS results are shown in the table below.
> > >
> > > |         | PSNR      | SSIM       | LPIPS      |
> > > | ------- | --------- | ---------- | ---------- |
> > > | Ficus   | 32.35     | 0.9773     | 0.0191     |
> > > | Hotdog  | 35.66     | 0.9806     | 0.0297     |
> > > | Lego    | 34.51     | 0.9766     | 0.0208     |
> > > | Mic     | 34.01     | 0.9871     | 0.0129     |
> > > | Average | **34.13** | **0.9804** | **0.0206** |
> > > | Ours    | 30.11     | 0.9466     | 0.0528     |
> > >
> > > Although the NVS results are significantly higher than ours, the decomposition of each component is completely chaotic, with shadows baked into the albedo, the environment map being meaningless, and so on. This issue arises because the geometric quality of 3D-GS is far inferior to that of NeuS.
> > >
> > > In the future, we will attempt to base our method on models like 2D-GS, which possess both good geometric and rendering quality. We believe that strong geometric quality is the foundation for the correct decomposition in inverse rendering, facilitating accurate BRDF estimation and yielding better PBR materials and environment maps for relighting and shadow removal. Improved rendering quality will also enhance NVS quality and further improve the rendering quality of relighting.

---

### Official Review · Reviewer_ySN8 · 2024-07-19

**Soundness:** 2
**Presentation:** 3
**Contribution:** 2
**Rating:** 6
**Confidence:** 5

**Summary:**

This paper introduces RobIR, an inverse rendering approach designed to handle strong or directional illumination scenes with strong shadows and specular reflections.
The proposed method aims to decouple environment lighting and object materials, with the goal of producing high-quality albedo without baked shadow.

Building on top of prior inverse rendering methods such as InvRender, RobIR introduces two components that further boost the reconstruction quality: (1) ACES tone mapping with an optimizable gamma parameter to better capture the image formation process; (2) regularization for visibility estimation.

RobIR demonstrates better performance over existing methods in quantitative and qualitative evaluations.

**Strengths:**

- The model design choices are valid and sensible.
- The experiments and ablation study are thorough.
- The paper is well written and easy to follow.

**Weaknesses:**

[W1] The benefit of ACES tone-mapping is a bit surprising. Despite a more accurate formulation for the image formation process, the task of inverse rendering is inherently still an ill-posed problem. It’s a surprising conclusion that a tone-mapping formulation can robustly and significantly benefit shadow removal.

With many other regularization terms entangled, it’s a bit hard to evaluate the correctness of this specific component.
I’d hope to know more details in the following aspects:


[W1.1] Missing visualization of the tone-mapping curve. Despite an important contribution, the estimation results of the tone-mapping curve are missing. What does the default ACES tonemapping look like, and what does the final tonemapping look like with the optimizable gamma?

In the revised version, the curves should be qualitatively visualized and included in main paper. From the existing experiments (such as the PSNR metrics), the audience cannot intuitively understand how well the tonemapping is estimated.

[W1.2] Missing evaluation of the tone-mapping curve.

As many datasets do not have GT tonemapping, it’s unclear how accurately the tone-mapping approximate the GT tonemapping.

One way to evaluate is to add additional tone adjustment to the input dataset. Assume with the original dataset images $\{ I \}$ and the method originally reconstructs a tonemapping curve $f$. Given a new tone adjustment function, e.g. $g(x) = x^\kappa$, the adjusted dataset images become $\{g(I)\}$. Re-running the method can get a newly reconstructed tonemapping curve $f_\kappa $. The consistency between $g \circ f$  and $f_\kappa$ can indicate how well the model can approximate the additional introduced tone adjustment function. $\kappa$ can be set to values like 0.5 or 2.

[W1.3] The evaluation metric on the Albedo is flawed. As albedo estimation/optimization often involve an unknown scale, PSNR alone is not a proper evaluation for Albedo. Check out [1] for more analysis and more appropriate scale-invariant metrics.

[W1.4] Most of the results are from synthetic datasets, where the GT tonemapping could potentially be close to ACES. For real-world results in Fig.7, the albedo estimation looks strongly regularized and over-smoothed.

[W1.5] Are the radiance values (before tonemapping) and indirect illumination in HDR? If so, what is the activation function?

[W2] The proposed method involves a complicated training pipeline (two stages with each stage have its own loss scheduling), and the novelty is relatively limited.


**References**

[1] Grosse et al., Ground truth dataset and baseline evaluations for intrinsic image algorithms, ICCV 2009.

**Questions:**

This paper addresses the shadow baking issue in inverse rendering by proposing two straightforward but effective techniques: optimizable tone-mapping and visibility regularization.

However, I am not fully convinced that optimizable tone-mapping significantly benefits shadow removal, and without further clarification it’s challenging to evaluate the technical correctness of this component. In the rebuttal, please prioritize addressing Weaknesses 1.1-1.4.

**[Post rebuttal update]**

The rebuttal and related discussion address my concerns. I update my rating to Weak Accept.

**Limitations:**

The paper discussed limitations in the end of paper. It’ll be beneficial to include failure cases in Appendix.

---

> ### Author Rebuttal · Authors · 2024-08-06
>
> We are glad and appreciate that the reviewer recognize that the results of RobIR is thorough. Our response to the reviewer’s comments is below:
>
> **Q1: Visualization of the tone-mapping curve.**
>
> Great suggestion. We highly appreciate the suggestion, which can improve the readability of the article. We have already submitted visualizations with different tone mappings and different gamma settings in the submitted rebuttal PDF.
>
>
>
> **Q2: Missing evaluation of the tone-mapping curve.**
>
> This is also a very good suggestion. We would like to thank the reviewer for the in-depth understanding and for attempting to provide a method to evaluate real-world scenarios without GT tone mapping. Here, we provide tone mapping evaluations for two our rendered scenes: truck and chessboard.
>
> - Truck: We rendered the scene in Blender and used Blender's built-in Filmic tone mapping to handle the high dynamic range. The final optimized $\gamma$ for the `truck` scene is 1.0, meaning the optimized tone mapping is the vanilla ACES. Considering that the Filmic curve is modified from ACES and their distributions are generally consistent with only some differences in detail, we believe that the optimization of tone mapping for this scene is accurate.
> - Chessboard: We also rendered this scene in Blender, but used sRGB to process the BRDF output. The final optimized $\gamma$ value is 0.42. In the submitted PDF, we compared the differences between the ACES curve with $\gamma=0.42$ and the sRGB curve. It can be seen that $\gamma^{-0.2}$ stretches the ACES curve very well, making it closer to the distribution of the sRGB curve. Combined with more accurate visibility modeling (NeuS Octree and Regularized Visibility Estimation), we can remove the baked shadows and reflections from the PBR material in the chessboard scene (See Fig. 6 and Fig. 8).
>
>
>
> **Q3: Albedo quantitative comparison.**
>
> We greatly appreciate the reviewer's attention to the hue differences in albedo in inverse rendering, as directly using PSNR might lead to unfair comparisons. We carefully studied the "Ground Truth Dataset and Baseline Evaluations for Intrinsic Image Algorithms" for evaluating and decomposing albedo and found that their decomposition formula is $I(x) = S(x)R(x)+C(x), \text{where}\  S(x), R(x), C(x)\  \text{denotes illumination, albedo, specular respectively} $, which is not compatible with our method. Nonetheless, we are grateful to the reviewer for suggesting a more reasonable evaluation of albedo.
>
>
>
> Regarding the evaluation of albedo, we believe that qualitative assessment is more important. Therefore, in Fig. 4, we provide extensive comparisons to demonstrate that our method does not bake shadows into the albedo. The removal of shadows cannot be solved merely by adjusting the hue (scale). Quantitative experiments only serve as a supplement to albedo evaluation. Although every method has issues with hue affecting PSNR to some extent, our method remains the closest and does not bake shadows. Furthermore, we also provide SSIM and LPIPS metrics, which are less sensitive to hue, and these metrics consistently show that our method achieves higher quality albedo, in agreement with Fig. 4.
>
> **Q4: Albedo estimation in real-world scenes looks over-smoothed.**
>
> In Fig. 7, the `Man` scene is actually successful in removing complex lighting and shadows, resulting in a smooth albedo, as the material of the sculpture is uniform. However, in the `Bear` scene, there is indeed an issue of over-smoothing. As mentioned in the common response, the smooth loss does have some impact, but it is more attributable to the rendering precision of NeuS. We are very much looking forward to considering structures that balance both rendering quality and geometric quality, such as 2DGS, in our future work.
>
>
>
> **Q5: About the radiance values and indirect illumination.**
>
> Your understanding is very accurate. The radiance values (before tone mapping) and indirect illumination are in HDR, which are then mapped to the $[0,1]$ using scene-specific ACES tone mapping. The activation functions for albedo and roughness are **sigmoid**, while SG-based direct light is modeled as `nn.Parameter`. The indirect light is supervised by NeuS (with NeuS supervision values $[0, 1]$ are mapped back to the same range in BRDF estimation stage using the inverse ACES mapping in Eq. 7).
>
> **Q6: The statement on the role of scene-specific ACES tone mapping.**
>
> In Figure 8, we conducted a rigorous ablation study on the effect of ACES. Introducing scene-specific tone mapping indeed plays a crucial role in shadow removal.
>
> Finally, we would like to thank the reviewer once again. Many of these points were things we had not noticed before, and these suggestions will significantly improve the readability of the paper.

---

### Author Rebuttal · Authors · 2024-08-05

We thank all the reviewers for their valuable comments. We are glad and appreciate that the reviewers recognize that our proposed regularized visibility estimation and ACES tone mapping are novel, and our experiments are thorough and impressive. We will further polish our paper and release our codes.



We would first like to clarify the contributions of RobIR and provide the common response for the explanation of details absence and ACES tone mapping. Following that, we will address the specific questions posed by each reviewer.



**1. Contribution:**

We are very grateful to reviewer `GRUR` for pointing out that we used the original NeRF rendering of Hotdog and Lego instead of NeRFactor’s. This actually highlights the fundamental contribution of our work. Currently, most neural field-based inverse rendering methods are applied to scenes with low light intensity (as shown in Fig. 13), which avoids the challenge of decoupling shadows, indirect illumination, and PBR materials. This reflects that the components decomposed by current inverse rendering methods are often messy and fail to decouple shadows from PBR materials in high illumination scenes (see Fig. 4).

Through this paper, we aim to draw the community's attention to this basic issue in current IR frameworks, and we propose our solution by explicitly introducing ACES tone mapping and more accurate visibility modeling.



**2. The absence of the texture details in Fig. 4 and Fig. 7.**

Reviewers `ijRb` and `GRUR` believe that our regularization terms may lead to a loss of high-frequency details. The loss of some details is indeed an issue brought about by smooth loss. Additionally, we believe that the rendering accuracy of NeuS is a more significant reason for the loss of details. However, we want to emphasize that **the quality of geometry** is crucial for correctly understanding the scene and decoupling materials and shadows. Only through NeuS's high-quality geometric reconstruction and smooth loss correction of geometric errors can we correctly decouple shadows from the object's PBR materials, ensuring that shadows or indirect illumination are not baked into the PBR materials (**No shadow residual** in Fig. 7 bear and Fig. 4 truck). We look forward to incorporating work that excels in both geometric reconstruction and rendering quality, such as 2DGS, in future endeavors to enhance the details of PBR materials.



**3. Why ACES can make a difference in BRDF estimation.**

We are very grateful that almost every reviewer raised this issue; it is a very profound question. We believe that the introduction of scene-specific ACES tone mapping has two benefits for BRDF estimation.

- First, as reviewer `ySN8`mentioned, incorporating ACES tone mapping undoubtedly provides a more accurate formulation for the image formation process, especially for high-illumination scenes (since rendering in such scenes often requires HDR tone mapping).
- Second, and what we consider a more direct reason in neural field-based inverse rendering, is that it offers a broader value range for the calculation of PBR color (Lines 159-163). In the submitted rebuttal PDF, we provided a comparison **between ACES and sRGB tone mapping**. It can be observed that the input for sRGB is within the range of $[0, 1]$, whereas ACES extends from $[0, +\infty]$. This allows the color obtained through PBR to have a greater contrast when aided by ACES tone mapping, thereby helping to decouple the shadows in extremely dark areas from the PBR material.

However, not all scenes with shadows or high illumination undergo HDR tone mapping, and there are differences between various HDR tone mapping methods (as reviewer `GRUR` mentioned, such as Filmic and AgX curves). Therefore, we introduced a $\gamma$ correction (see Eq. 8) to stretch the ACES curve, allowing it to adapt to different light intensities and to approximate various tone mapping curves. We also included a comparison of curves under different $\gamma$ values with other tone mapping methods in the submitted rebuttal PDF.



Overall, we believe that the introduction of scene-specific ACES tone mapping undoubtedly provides a foundation for more precise BRDF estimation. Its inclusion is essential for the decoupling of shadows, and when combined with our proposed Regularized Visibility Estimation (RVE), it enables more robust shadow removal. As shown in the Fig. 8 ablation study, neither ACES nor RVE alone can achieve reliable shadow removal, but together they can truly make a difference.

---

### Decision · Program_Chairs · 2024-09-25

**Decision:**

Accept (poster)

**Comment:**

The paper presents a new approach to inverse rendering, specifically designed for challenging high-illumination scenarios. The method introduces scene-specific tone mapping and regularized visibility estimation to effectively decouple environment lighting from an object's BRDF, resulting in improved albedo and roughness estimations without shadow interference. The paper is well-written and supported by comprehensive experiments, demonstrating superior performance over existing methods.

While the proposed approach is novel, some reviewers noted that it builds incrementally on existing frameworks like NeuS and InvRender, and the complexity of the method may limit its practical application. Additionally, there were concerns about the loss of high-frequency details due to the smooth loss used in the model, which the authors addressed by attributing it to the rendering precision of NeuS. Despite these concerns, the authors provided thorough rebuttals that clarified the benefits of their approach, particularly the advantages of ACES tone mapping in HDR contexts.

After the discussions, some reviewers raised their initial scores, recognizing the paper's contributions. Given the strengths of the work and the constructive engagement with reviewers, the paper is recommended for acceptance as a poster presentation.